# Hepatitis of Unknown Origin and Etiology (Acute Non HepA-E Hepatitis) among Children in 2021/2022: Review of the Current Findings

**DOI:** 10.3390/healthcare10060973

**Published:** 2022-05-24

**Authors:** Malik Sallam, Azmi Mahafzah, Gülşen Özkaya Şahin

**Affiliations:** 1Department of Pathology, Microbiology and Forensic Medicine, School of Medicine, The University of Jordan, Amman 11942, Jordan; mahafzaa@ju.edu.jo; 2Department of Clinical Laboratories and Forensic Medicine, Jordan University Hospital, Amman 11942, Jordan; 3Department of Translational Medicine, Faculty of Medicine, Lund University, 22362 Malmö, Sweden; gulsen.ozkaya_sahin@med.lu.se; 4Laboratory Medicine, Department of Clinical Microbiology, Skåne University Hospital, 22242 Lund, Sweden

**Keywords:** outbreak, novel, unknown hepatitis, unknown etiology, public health promotion

## Abstract

Several clusters and individual cases of acute hepatitis have been reported in the US, Europe and recently in Asia and Central America since October 2021. A laboratory investigation of the common viral hepatitis agents (HAV, HBV, HCV, HDV and HEV) yielded negative results prompting the use of the term “acute non HepA-E hepatitis” to describe this condition. The cases were characterized by the manifestations of acute hepatitis (abdominal pain, vomiting, diarrhea, jaundice and very high levels of liver enzymes) affecting children with a median age of 3–4 years. The exact underlying etiology has not been revealed yet; however, a leading hypothesis is that an infectious agent is the culprit, underlying cause or a risk factor for acute non HepA-E hepatitis occurrence. So far, laboratory testing has shown the presence of the group F human adenovirus serotype 41 (HAdV-F41) in about three-fourths of the investigated cases. As of 13 May 2022, more than 450 cases were reported worldwide, the majority of which were in the UK (*n* = 176), the US (*n* = 109), 13 European countries (at least 103 cases) and in Argentina, Brazil, Canada, Costa Rica, Indonesia, Israel, Japan, Palestine, Panama, Singapore and South Korea. Vigilant surveillance and epidemiologic investigations to identify further cases are warranted to delineate the features of this emergent public health issue. The possible role of environmental and toxic agents including foodborne toxins should also be considered. Specific guidelines for identification of further cases are necessary, particularly in low-income settings where testing for adenoviruses is not considered routinely. A genetic analysis of HAdV-F41 isolates is recommended to assess the potential changes in the virus genome with subsequent possible altered virus behavior. Immunopathogenesis is another possibility that should be evaluated considering the lack of viral structures in liver biopsies of the affected children in the US.

## 1. Introduction

An ongoing investigation has started since January 2022 to evaluate a higher than usual number of hepatitis cases in children, the majority of which took place in the United Kingdom (UK) [1,2]. As of the end of April 2022, cases were reported in three different continents (Europe, North America and Asia) with a high concentration of cases in Europe [1,3,4,5,6]. These cases affected children aged 16 years and younger with a predilection for those aged one to five years [4,5,6,7]. The absence of any link between these cases collectively termed “acute non HepA-E hepatitis” and the currently known viral hepatitis agents (HAV, HBV, HCV, HDV and HEV) prompted the investigation of this emergent condition to decipher its possible etiology, pathogenesis and outcome [4,5,6,8].

One of the alarming features of acute non HepA-E hepatitis in children is the unusually high proportion of severe cases that necessitated liver transplantation in a fraction of the affected cases [4,6]. For example, out of 163 cases that were confirmed in the UK up until 3 May 2022, 11 cases required liver transplantation [9]. Similarly, two out of nine cases in Alabama, US, required the similar management option for severe hepatitis [3,7]. Moreover, several cases of mortality were attributed to acute non HepA-E hepatitis in the US (five deaths), Indonesia (three deaths), Ireland and Palestine (one death in each country), which display the threatening potential of this condition [6,10,11,12,13,14].

The observation that the reported cases have been severe so far might hint to the possibility that the current situation represents the tip of an iceberg. This entails a scenario where a higher number of unreported mild or clinically inapparent cases have already been present; nevertheless, they have been unnoticed due to the absence of clinical features. The second ominous scenario encompasses the genuine severity of such an enigmatic clinical condition [15].

The examination of the possible etiology of acute non HepA-E hepatitis revolves around the following working hypotheses: (1) cofactors rendering a common and mild infection into a more severe form; (2) the role of a previously known infectious agent undergoing a critical genetic shift changing its pathogenicity; (3) co-infection between two different viruses; (4) the emergence of a novel infectious agent; (5) the possible role of immunopathology following an infection; and (6) the potential role of toxic or environmental agents including foodborne toxins [4,8].

The observation of adenovirus’ common presence in a majority of cases that have been identified and tested so far might point to the possible role of this virus in acute non HepA-E hepatitis [4,6,7,8]. This observation is interesting and unique due to the previous scarcity of reports pointing to adenovirus’s causal role in hepatitis in immunocompetent children [16,17,18]. In addition, the scarce reports on the role of adenovirus in hepatitis among the immunocompromised individuals pointed to the role of group C viruses, belonging to serotypes 1, 2 and 5, rather than adenovirus serotype 41 (HAdV-F41) classified as a group F adenovirus, which was found in the recently reported cases [7,19,20].

In this narrative review, we aimed to provide an overview of the current status of these unusual cases of hepatitis of unknown origin among children tackling the issues of: (1) case definition, (2) current geographic distribution, (3) clinical features, (4) possible hypotheses of etiology and (5) laboratory investigation. Herein, we opted to adopt the nomenclature by the World Health Organization (WHO) and the UK Health Security Agency using the term “acute non HepA-E hepatitis” to refer to hepatitis of unknown etiology/origin in children [6,8].

## 2. Case Definition for Acute Non HepA-E Hepatitis

The current working case definitions of confirmed, probable, possible and epidemiologically linked cases for surveillance are provided by the WHO, the European Centre for Disease Prevention and Control (ECDC) and the UK Health Security Agency (Table 1) [4,6,8]. The case definition is based on age, time of presentation, clinical presentation including levels of liver enzymes and the absence of markers of acute viral hepatitis excluding the presence of acute HAV, HBV, HCV, HEV and HDV among those with chronic HBV infection.

Currently, the WHO does not provide a definition for a confirmed case of acute non HepA-E hepatitis, likely due to the absence of a clear-cut definition of the underlying cause and the possible presence of other infectious and non-infectious agents that can give a similar clinical picture [6]. Similarly, the joint ECDC/WHO case definition for a confirmed case is not applicable at the present time [4]. However, the earlier ECDC guidelines and the UK Health Security Agency define a confirmed case of acute non HepA-E hepatitis based on the presentation of acute hepatitis with elevated level(s) of transaminases (alanine aminotransferase (ALT) or aspartate aminotransferase (AST)) higher than 500 international units (IU)/L in children aged 10 years or less any time from 1 January 2022 [4].

The WHO currently defines a probable case of acute non HepA-E hepatitis based on the presence of acute hepatitis, with elevated level(s) of ALT or AST higher than 500 IU/L, among those aged 16 years or younger, since 1 October 2021 [6]. The epi-linked cases based on the WHO definition rely on the presentation among individuals of any age who have been in a close contact with a probable case [6].

Possible cases based on the earlier ECDC definition and the UK Health Security Agency definition rely on the presence of acute hepatitis, with elevated level(s) of ALT or AST higher than 500 IU/L, among those aged between 11 and 16 years, since 1 January 2022, with epi-linked cases defined as close contacts of a confirmed case since the New Year 2022 [4,8].

## 3. Current Geographic Distribution of the Reported Acute Non HepA-E Hepatitis Cases

The geographic distribution of acute non HepA-E hepatitis cases is illustrated in Figure 1 with a detailed description in Table 2.

The first acute non HepA-E hepatitis cases were reported in Alabama, the US, where nine cases were identified between October 2021 and February 2022 [3]. A recent report showed that all the nine cases came from geographically distinct parts of Alabama without any epidemiologic linkages among the detected cases [7]. By 5 May 2022, 109 cases were reported in the US across 24 different states (Figure 2) [11]. In Canada, seven cases were identified between 1 October 2021 and 30 April 2022 [21].

In the UK, cases were reported from January 2022 to 10 May 2022 with a total of 176 cases [22]. As of 3 May 2022, the majority of these cases were detected in England (*n* = 118), followed by Scotland (*n* = 22), Wales (*n* = 13) and Northern Ireland (*n* = 10) [6,8,9].

The most comprehensive report on these cases came from Scotland [5], which pointed to an absence of clear connection between the reported cases, despite the identification of the condition in close contacts of two cases in the country [5,23].

In other European countries, recent reports from Italy pointed to the detection of 35 cases from March 2022 to 13 May 2022 [4]. Further cases in Europe as reported by the ECDC were distributed as follows: Spain (*n* = 22), Sweden (*n* = 9), Portugal (*n* = 8), Denmark (*n* = 6), Ireland (*n* = 6) and the Netherlands (*n* = 6), with cases in an additional ten European countries (Figure 3) [4,12,24].

On 25 April 2022, Japan’s health ministry reported a case of acute non A-E hepatitis, the first to be reported in Asia [25,26]. Six more cases were reported in the country, with similar presentation among individuals younger than 16 years without further details from Japan’s health ministry [27]. Indonesia reported a total of 15 cases with three mortalities in children as a result of the disease [13].

By the end of April 2022, an infant negative for the common hepatitis viruses was being investigated for acute non A-E hepatitis in Singapore, with a report confirming the condition in a 10-month-old baby [28,29]. Additionally, cases of acute non A-E hepatitis were identified in South Korea and Malaysia [30,31].

In Middle and South America, twenty-eight cases were reported in Brazil, eight cases in Argentina, two cases in Costa Rica and a single case in Panama by 13 May 2022 [32,33,34,35].

In the Middle East, Israel and Palestine were the only countries with cases of acute non HepA-E hepatitis up to 13 May 2022. In Israel, 12 cases of hepatitis of unknown origin were reported to the Ministry of Health as of April 2022 [36]. In Palestine, an 8-year-old child died as a result of the disease in the Gaza Strip in early May 2022 [14].

**Table 2 healthcare-10-00973-t002:** Detailed description of the acute non HepA-E hepatitis cases per country as of 13 May 2022.

Region/Country	Number of Acute Non HepA-E Hepatitis Cases	Notes
**North America**		
United States	109	Five cases of death were reported. Sources in [11,37]
Canada	7	Source in [21]
**Europe ***		
United Kingdom	176	Source in [22]
Italy	35	Source in [24]
Spain	22	Source in [24]
Sweden	9	Source in [24]
Portugal	8	Source in [24]
Denmark	6	Source in [24]
Ireland	6	Age range was 1–12 year, one died and once received liver transplantation. Source in [12]
Netherlands	6	Source in [24]
Norway	4	Source in [24]
Belgium	3	Source in [24]
Austria	2	Source in [24]
Cyprus	2	Source in [24]
France	2	Source in [24]
Germany	1	Source in [4]
Poland	1	Source in [24]
Romania	1	Source in [24]
Serbia	1	Source in [24]
Slovenia	1	Source in [38]
**Central & South America**		
Brazil	28	Source in [32]
Argentina	8	Source in [35]
Costa Rica	2	Source in [33]
Panama	1	Source in [34]
**Asia**		
Indonesia	15	Three cases of death were reported in children aged 2, 8 and 11. Source in [13]
Japan	7	
Malaysia	1	Source in [31]
Singapore	1	Source in [29]
South Korea	1	Source in [30]
**Middle East**		
Israel	12	Source in [36]
Palestine	1	One case of death in a child aged 8 years in Gaza. Source in [14]

* Two additional cases were reported in Greece [39].

## 4. Clinical Presentation and Severity

So far, the main affected group by acute non HepA-E hepatitis has been the previously healthy children, aged 16 years or younger with an absence of comorbidities [3,4,7]. The median age was three and four years both in the US and Scotland, respectively [5,7]. A majority of affected children were 10 years old or younger. Specifically, out of the 13 cases reported in Scotland, 12 were 5 years or younger in age [5]. Six out of nine patients in Alabama, US were younger than 5 years [7].

The predominant symptoms included abdominal pain, vomiting and diarrhea reported in the preceding weeks of hospital admissions [6]. In addition, very high levels of liver enzymes (ALT and AST) were found along with jaundice [5,6]. A striking feature is the very high levels of serum aminotransferases exceeding 500 IU/L. Specifically, Julia M. Baker et al. reported that the ALT level ranged between 603 and 4696 IU/L among the nine affected children in Alabama, while the range for AST was between 447 and 4000 IU/L [7].

In most cases, fever was absent as reported by the WHO and the ECDC [4,6]. The same pattern was reported in the cases from Scotland, where no fever was reported in the few weeks prior to hospital admission [5]. However, the report on Alabama cases displayed the presence of fever in 5/9 (55.6%) of the cases [7].

In the Alabama cases, upper respiratory tract symptoms were reported in one-third of the children prior to their admission [7]. Vomiting and diarrhea were reported by more than two-thirds of the cases both in Alabama and in Scotland [5,7]. In the Alabama cases, seven patients had hepatomegaly, and one had encephalopathy on admission, with seven patients recovering without liver transplantation compared to two cases that recovered following transplantation [7], showing the severity of acute non A-E hepatitis.

## 5. Hypotheses of Possible Etiology

### 5.1. Adenovirus Role

Currently, the most plausible hypothesis to explain cases of acute non HepA-E hepatitis among children entails the role of adenovirus infection [4,6,8]. In the technical briefing by the UK Health Security Agency, the working hypothesis with the best fit to surveillance data assumes that a normal adenovirus infection in children can be complicated by a cofactor that renders this infection into a severe form or can trigger immunopathology [8]. Possible cofactors include the issue of higher susceptibility as a result of lower exposure to adenoviruses during the coronavirus disease 2019 (COVID-19) pandemic, with the widespread adoption of non-pharmaceutical interventions and subsequent reduction of exposure to various pathogens [40]. Therefore, the restrictions imposed amid the ongoing COVID-19 pandemic may have led to later exposure of young children to adenoviruses, with delayed exposure resulting in a more vigorous immune response causing severe hepatic damage [8]. Such a scenario was suggested by Ruben H de Kleine et al. who reported a preliminary absence of a notable increase in pediatric acute liver failure upon comparing data covering 2019–2021 to those in the first 4 months of 2022 [41].

Other possible cofactors include prior or co-infection by other viruses including severe acute respiratory syndrome coronavirus 2 (SARS-CoV-2) or the exposure to toxins or other environmental agents [4,8].

Another possible hypothesis to explain acute non HepA-E hepatitis includes the role of adenovirus as well; however, this hypothesis assumes that a novel variant of adenoviruses is the underlying cause with or without the aforementioned cofactors [4,8].

The argument in favor of the adenovirus role in acute non HepA-E hepatitis is based on the observation that more than three-fourths of the reported cases were positive for adenoviruses [4,6,8]. Scientists and clinicians are recommended to investigate whether there has been a change in the genome of the virus that might allow for hepatotropism, triggering severe liver inflammation.

One important point that should be evaluated involves the reporting of acute non HepA-E hepatitis amid the COVID-19 pandemic. This pandemic was accompanied by enhanced molecular testing capacity for viruses worldwide; thus, the detection of previously unknown cases of hepatitis linked to HAdV-F41 aggravated by delayed exposure to this common infection might be a plausible hypothesis that requires further confirmation.

### 5.2. COVID-19

One of the hypotheses currently under consideration regarding the origin of acute non HepA-E hepatitis is the possible role of SARS-CoV-2 infection [8]. Specifically, a new variant of this virus can have a possible role. However, such a hypothesis does not appear likely, considering the absence of SARS-CoV-2 positivity (current or previous) in a majority of cases. Specifically, eight out of the thirteen cases reported in Scotland were negative for SARS-CoV-2 by polymerase chain reaction (PCR) testing, and an additional two cases had a positive testing result 3 months or earlier prior to admission [5]. Furthermore, all nine Alabama cases tested negative for SARS-CoV-2, minimizing the likelihood of a direct role of COVID-19 in acute non A-E hepatitis [7].

Moreover, acute hepatitis has not been a common feature of COVID-19 in children, despite the presence of reports on its potential occurrence [42].

Variants that escape the currently available testing modalities for molecular detection appear unlikely as well, considering the previous evidence showing that the diagnostic accuracy of the PCR was not impacted by emerging SARS-CoV-2 variants, including the most recent dominant genetic lineage, namely, omicron [43].

### 5.3. COVID-19 Vaccination

Even though the possibility of a link between acute non A-E hepatitis cases and side effects following COVID-19 vaccination was considered, such a hypothesis appears remote and unreasonable considering that the majority of cases occurred in children that were not vaccinated against COVID-19 [6,44]. As the current reports, albeit scarce, pointed to the predominance of acute non A-E hepatitis in young children—with children 5 years or younger not eligible for COVID-19 vaccination—this observation almost excludes the possibility of COVID-19 vaccination’s role in this emerging issue [45].

### 5.4. Novel Infectious Agent Etiology

The potential role of an emerging novel infectious agent should be considered, with viruses on top of the list. The RNA viruses appear as the primary candidates, considering the relatively swift evolution of such viruses with the subsequent ability of successful cross-species transmission, adaptation to new niches and a rapid change in virulence [46].

### 5.5. Toxic Agent

The preliminary investigation of a possible link between acute non A-E hepatitis cases in the UK and potential toxic agents (e.g., metals in urine, organic compounds in serum) did not reveal any significant findings upon comparing the cases (*n* = 11) and healthy controls (*n* = 16) [8]. This may hint to a lower possibility of a link between hepatotoxic agents and the current cases [4]. However, the toxicologic investigation is ongoing, and the exclusion of a possible causal role of toxic agents is pending further evidence [8].

### 5.6. Foodborne Etiology

A recent report by the European Society of Clinical Microbiology and Infectious Diseases (ESCMID) raised a relevant point in relation to the possibility that the current cases might be related to a foodborne outbreak. This hypothesis should be tested considering that food production can be a centralized process with distribution from a single manufacturer to different destinations worldwide [47]. Aflatoxins emerge as the primary candidate for involvement in this case scenario, considering their association with acute severe hepatic damage among the exposed individuals [48].

## 6. Laboratory Investigation

Based on case definition and the aforementioned hypotheses regarding the possible origin of acute non HepA-E hepatitis, serology testing for HAV-HEV should be conducted among the suspected cases [4,8]. Molecular testing for adenoviruses displayed a higher yield in whole blood compared to plasma samples, which should be considered in testing as well [7]. The detection of adenovirus in the blood in spite of a relatively low viral load provides a hint to the possibility of disseminated infection [4].

Notably, Epstein Barr virus (EBV) was identified using molecular detection among 6/9 cases in Alabama, US; however, the absence of IgM antibodies suggests that these patients were experiencing reactivation of EBV rather than a primary infection [7].

Despite the observation that a minority of cases were positive for SARS-CoV-2, molecular testing has been advocated among suspected cases of acute non HepA-E hepatitis [49].

The comprehensive report by Julia M. Baker et al. reported the results of liver biopsies from six children with acute non HepA-E hepatitis which displayed the absence of viral inclusions, and the complete lack of an adenovirus presence by electron microscopy and immunohistochemical staining [7]. Further testing of liver biopsies from severe cases can give more in-depth insights regarding the proposed role of adenoviruses and their subsequent immunopathology in these cases.

The current guidelines for laboratory testing in the suspected cases of acute non HepA-E hepatitis include: (1) PCR testing using blood/serum samples for adenoviruses, enteroviruses, human herpes viruses 1, 2, 3, 4, 5, 6 and 7, hepatitis A, C and E viruses; (2) serologic testing for hepatitis A, B, C and E viruses, EBV and cytomegalovirus (CMV) besides SARS-CoV-2; (3) blood culture for bacteria if fever is present; (4) multiplex PCR respiratory viruses panel (that includes adenoviruses, enteroviruses, influenza viruses, human bocaviruses and SARS-CoV-2) on the earliest throat swab possible; (5) multiplex PCR gastrointestinal viruses panel (that includes adenoviruses, sapovirus, norovirus, enteroviruses) on a stool sample; (6) stool culture for common bacterial enteropathogens including *Salmonella* spp. [4,49]. Serologic testing for anti-streptolysin O (ASO), throat swab culture for group A β-hemolytic *Streptococci* and serum/urine testing for leptospirosis should be considered if clinically indicated. Toxicologic screening using blood and urine samples should be considered as well [49].

## 7. Discussion

The accelerated increase in severe acute hepatitis cases in children —currently termed acute non HepA-E hepatitis— that was reported in 19 countries worldwide is worrisome and requires vigilant surveillance and coordinated efforts to identify the potential underlying etiology, possible transmission routes and origin if an infectious agent was found to be implicated in this emergent public health problem.

The identification of further cases in different countries is critical primarily in the countries where cases were first recorded, as well as in other countries worldwide. Thus, surveillance for such cases is of prime importance. This should be followed by the investigation of possible epidemiologic links. The report by Kimberly Marsh et al. from Scotland showed that two children had close contact in a household or other setting with two other cases, in spite of the absence of any known epidemiologic linkages in Alabama cases so far [5,7]. Therefore, a continuous epidemiologic investigation is warranted to resolve such a discrepancy and to help in the identification of possible mild or inapparent cases. Subsequently, this approach can be helpful to determine the underlying etiology of acute non HepA-E hepatitis.

Another aspect of the possible infectious agent(s) involvement in acute non HepA-E hepatitis entails examining the hypothesis of the potential role of immunopathologic mechanisms. This appears as a plausible hypothesis, given the absence of adenoviruses in the liver biopsies examined in Alabama cases, using accurate detection methods including electron microscopy and immunohistochemical staining [7].

The routine testing for adenoviruses in the blood and stool samples is not considered routinely in the clinical practice, especially in countries with a limited molecular testing capacity. In turn, this may hinder the initial detection of cases if adenoviruses are truly implicated in acute non HepA-E hepatitis. Thus, it is critical to provide updated specific guidelines for adenovirus testing and reporting of cases based on clinical data or the establishment of simple, yet accurate, methods for adenovirus testing including lateral flow assays.

Another important issue to be considered based on the currently available evidence involving adenovirus’ role is the type of sample to be tested. It appears that molecular testing for adenoviruses among the suspected cases of acute non HepA-E hepatitis is better completed using whole blood rather than serum/plasma samples, considering the higher yield of whole blood for adenovirus detection [4,7].

If adenoviruses were genuinely implicated in acute non HepA-E hepatitis in children, several infection control measures would be necessary including the practices of proper hand hygiene and frequent surface disinfection [50]. This is related to the non-enveloped nature of adenoviruses, which are expected to be stable for long periods [51].

Although the role of infectious agents in acute non HepA-E hepatitis appears more sound taking into account the clinical features as well as the higher than usual prevalence of adenoviruses among the reported cases, other possible etiologies cannot be ruled out and should be examined meticulously, including the possible role of environmental agents, toxins or foodborne etiologies. Since the infectious etiology of acute non HepA-E hepatitis has not been confirmed yet, the toxicologic investigation and continuous quest for possible environmental causes should be considered among the top priorities.

Finally, we stress the importance of epidemiologic studies that are much needed to delineate the scope of discrepancy regarding the presence of epidemiologic links in the currently reported cases, which might be related to a genuine absence of such links or underreporting if the current cases represent the tip of an iceberg with much higher numbers of asymptomatic and mild cases.

## Figures and Tables

**Figure 1 healthcare-10-00973-f001:**
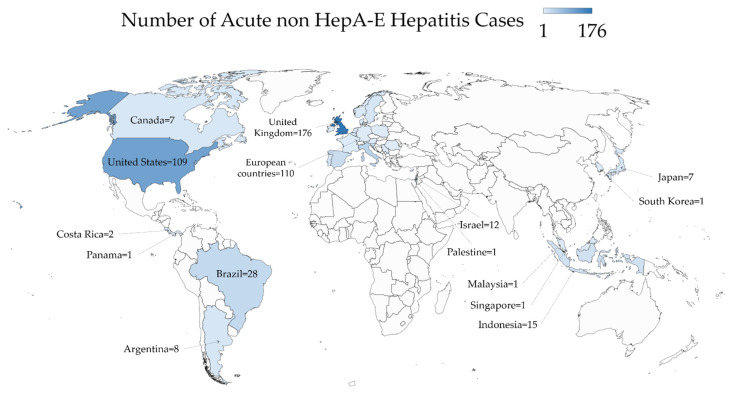
The global geographic distribution of acute non HepA-E hepatitis as of 13 May 2022. The map was generated in Microsoft Excel, powered by Bing, © GeoNames, Microsoft, Navinfo, TomTom, Wikipedia. We are neutral with regard to jurisdictional claims in this map.

**Figure 2 healthcare-10-00973-f002:**
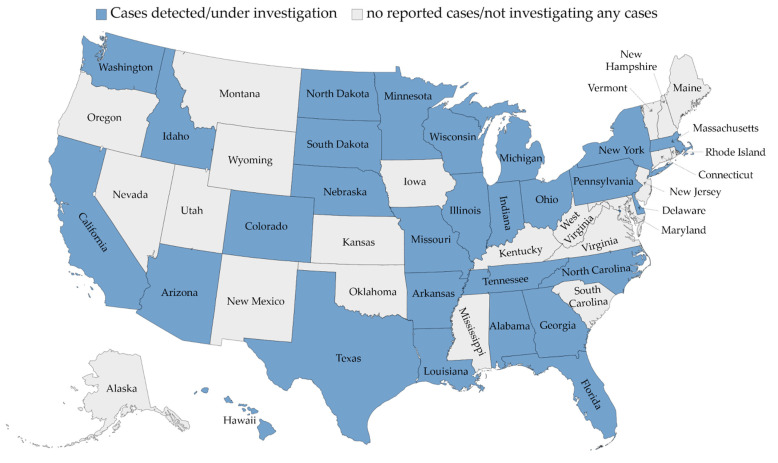
The distribution of acute non HepA-E hepatitis in the United States as of 13 May 2022. The map was generated in Microsoft Excel, powered by Bing, © GeoNames, Microsoft, Navinfo, TomTom, Wikipedia.

**Figure 3 healthcare-10-00973-f003:**
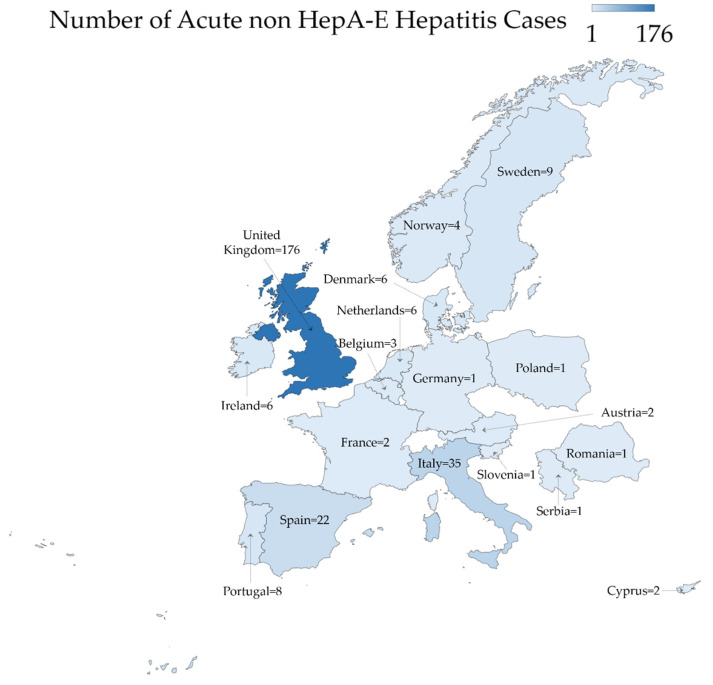
The geographic distribution of acute non HepA-E hepatitis in Europe as of 13 May 2022. The map was generated in Microsoft Excel, powered by Bing, © GeoNames, Microsoft, Navinfo, TomTom, Wikipedia. We are neutral with regard to jurisdictional claims in this map.

**Table 1 healthcare-10-00973-t001:** Working case definition of acute non HepA-E hepatitis.

Source	Case Definition
**WHO/ECDC**	
Confirmed acute non HepA-E hepatitis	Not applicable
Probable acute non HepA-E hepatitis	The presence of acute hepatitis with elevated level(s) of ALT or AST higher than 500 IU/L, and negativity for viral hepatitis (A–E) in children aged 16 years or younger, since 1 October 2021
Epi-linked acute non HepA-E hepatitis	The presence of acute hepatitis in a person who has been in close contact with a probable case of acute non HepA-E hepatitis, since 1 October 2021
**UK Health Security Agency**	
Confirmed acute non HepA-E hepatitis	The presence of acute hepatitis with elevated level(s) of ALT or AST higher than 500 IU/L, and negativity for viral hepatitis (A–E) in children aged 10 years or less any time from 1 January 2022
Possible acute non HepA-E hepatitis	The presence of acute hepatitis with elevated level(s) of ALT or AST higher than 500 IU/L, and negativity for viral hepatitis (A-E) in those aged between 11 and 16 years any time from 1 January 2022
Epi-linked acute non HepA-E hepatitis	The presence of acute hepatitis in a person who has been in close contact with a probable case of acute non HepA-E hepatitis, any time from 1 January 2022

Abbreviations: WHO: World Health Organization; ECDC: The European Centre for Disease Prevention and Control; UK: United Kingdom; ALT: alanine aminotransferase; AST: aspartate aminotransferase; IU: international unit.

## Data Availability

Data supporting this review are available in the reference section.

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
