# Peer review of "Hepatitis of Unknown Origin and Etiology (Acute Non HepA-E Hepatitis) among Children in 2021/2022: Review of the Current Findings"

_healthcare, 2022, doi:10.3390/healthcare10060973_

Round 1

Reviewer 1 Report

Overview and general recommendation:

Sallam and colleagues comprehensively compiled current findings regarding the recently emerged acute hepatitis among children around the world. Overall, the topic is very important and interesting, although the causative agent of this acute hepatitis is still unknown, and its origin and transmission routes remain to be determined. I have the following comments for the authors’ consideration.

  1. The authors used the term either “acute non hepA–E hepatitis” or “non A-E hepatitis” in the manuscript. Please unify them. How about using “non-A-E hepatitis”? In addition, please unify “etiology” and “aetiology”.
  2. The Abstract contains approximately 370 words, which is too much. Please consider deleting some redundant information. The reviewer checked that the maximum number of words for Abstract in this Journal is 200 words.
  3. Abstract, lines 30-31: why are there no exact case numbers for the Netherlands, Austria, Belgium, France, Germany, and Poland? Line 35: delete “as well”, line 36: change “is” to “are”.
  4. Section 3: I suggest moving lines 115-117 to the seventh paragraph, which describes the reported cases in the U.S. Lines 126-127: this sentence should be moved to the end of this paragraph since Israel is not a European country.
  5. In Figure 1, some countries’ colors are too shallow to read. For a better presentation, I suggest indicating the country names or three-digit codes for each of the countries with reported cases. Moreover, I suggest omitting Figure 3, whose information has already been shown in Figure 1. Otherwise, please provide necessary explanations to keep it.
  6. A tabulated format should be advantageous to summarize the current reported cases in different countries.
  7. Section 5, lines 190-193 and 204-207 seem highly similar, which can be combined and shortened.
  8. An additional Section including how to prevent this acute hepatitis of unknown etiology is recommended. For example, since the authors stated that the adenovirus infections are the most plausible hypothesis (lines 185-186), please specify some prevention actions against adenovirus infection.
  9. Please refer to the latest report from UK Health Security and Agency and update the case numbers through the manuscript accordingly. https://assets.publishing.service.gov.uk/government/uploads/system/uploads/attachment_data/file/1073704/acute-hepatitis-technical-briefing-2.pdf

Author Response

Reviewer #1 Comments and Suggestions for Authors

Overview and general recommendation:

Sallam and colleagues comprehensively compiled current findings regarding the recently emerged acute hepatitis among children around the world. Overall, the topic is very important and interesting, although the causative agent of this acute hepatitis is still unknown, and its origin and transmission routes remain to be determined.

Response: We are deeply thankful for the summary of the manuscript topic.

I have the following comments for the authors’ consideration.

  1. The authors used the term either “acute non hepA–E hepatitis” or “non A-E hepatitis” in the manuscript. Please unify them. How about using “non-A-E hepatitis”? In addition, please unify “etiology” and “aetiology”.

Response: We would like to thank the reviewer for this important remark, and accordingly we unified the referral to the condition as “acute non HepA-E hepatitis”

  1. The Abstract contains approximately 370 words, which is too much. Please consider deleting some redundant information. The reviewer checked that the maximum number of words for Abstract in this Journal is 200 words.

Response: We would like to thank the reviewer for this point. To meet the journal guidelines, we revised the Abstract to make it more concise.

  1. Abstract, lines 30-31: why are there no exact case numbers for the Netherlands, Austria, Belgium, France, Germany, and Poland? Line 35: delete “as well”, line 36: change “is” to “are”.

Response: In the original abstract prior to revision, we referred to the countries with similar number of cases as one statement “Netherlands and Ireland (n=4)” where the number in each country was 4. The same applies for “Austria, Belgium, France, and Norway (n=2)”, where the number of cases in each country was 2, and for “Germany, Poland, and Romania (n=1)” where the number of cases in each country was 1. Although we agree with the reviewer in this point, our response to the previous point, with extensive revision of the Abstract made us unable to accommodate the suggested change. We also would like to thank the reviewer for pointing to the grammatical error, and accordingly we corrected the statement as follows: line 39 of the revised highlighted manuscript: “Specific guidelines for identification of further cases are necessary …”

  1. Section 3: I suggest moving lines 115-117 to the seventh paragraph, which describes the reported cases in the U.S. Lines 126-127: this sentence should be moved to the end of this paragraph since Israel is not a European country.

Response: We would like to thank the reviewer for this important suggestion. We aimed to illustrate that the first cases were reported in the US in the original manuscript. However, based on the reviewer comment and based on the currently available data on the reported number of cases, we revised the entire section to make it easier to follow. Please refer to lines 131-171 in the revised highlighted manuscript.

  1. In Figure 1, some countries’ colors are too shallow to read. For a better presentation, I suggest indicating the country names or three-digit codes for each of the countries with reported cases. Moreover, I suggest omitting Figure 3, whose information has already been shown in Figure 1. Otherwise, please provide necessary explanations to keep it.

Response: We would like to thank the reviewer for raising this important point. Based on the suggestion, all figures were revised with addition of the country names and number of cases in each country. The justification of highlighting Europe in a single Figure is that many countries of the continent reported cases of acute non A-E hepatitis besides the difficulty to present such cases in the global map.

  1. A tabulated format should be advantageous to summarize the current reported cases in different countries.

Response: We would like to thank the reviewer for this important suggestion. Accordingly, we inserted a new Table (Table 2) to summarize the latest number of cases in each country based on the most up-to-date references.

  1. Section 5, lines 190-193 and 204-207 seem highly similar, which can be combined and shortened.

Response: We are thankful for this suggestion and based on the reviewer comment, we combined the two paragraphs.

  1. An additional Section including how to prevent this acute hepatitis of unknown etiology is recommended. For example, since the authors stated that the adenovirus infections are the most plausible hypothesis (lines 185-186), please specify some prevention actions against adenovirus infection.

Response: We are thankful for this important suggestion. However, due to absence of hard evidence suggestion that adenoviruses are truly implicated in acute non A-E hepatitis, we were inclined to add it to the discussion as follows: “If adenoviruses were genuinely implicated in acute non HepA-E hepatitis in children, several infection control measures would be necessary including the practices of proper hand hygiene and frequent surface disinfection [50]. This is related to the non-enveloped nature of adenoviruses, which are expected to be stable for long periods [51]”.

  1. Please refer to the latest report from UK Health Security and Agency and update the case numbers through the manuscript accordingly. https://assets.publishing.service.gov.uk/government/uploads/system/uploads/attachment_data/file/1073704/acute-hepatitis-technical-briefing-2.pdf.

Response: We would like to thank the reviewer for this important suggestion and accordingly we added the suggested reference: Reference No. 9 in the revised highlighted manuscript.

Reviewer 2 Report

This is a clearly written and well documented discussion of the current non-hepatitis A-E outbreak observed in in the UK, Europe and US. The etiology appears linked to human adenovirus F41 and the authors discuss various cofactors that may contribute to increased severity of Ad41 infection in these cases. The manuscript is accessible to a diverse audience. I only have two minor points regarding the submission:

  1. Current accepted nomenclature for adenoviruses adopts the format in this case as HAdV-F41 for Human Adenovirus species F strain 41. I think it would be worthwhile to use this nomenclature in this text.
  2. The color designations in Fig 2 in the UK region do not match the corresponding colors in this region in Figs 1 and 3. This should be corrected.

Author Response

Reviewer #2 Comments and Suggestions for Authors

This is a clearly written and well documented discussion of the current non-hepatitis A-E outbreak observed in in the UK, Europe and US. The etiology appears linked to human adenovirus F41 and the authors discuss various cofactors that may contribute to increased severity of Ad41 infection in these cases.

Response: We are deeply thankful for the concise summary of the review topic.

The manuscript is accessible to a diverse audience. I only have two minor points regarding the submission:

  1. Current accepted nomenclature for adenoviruses adopts the format in this case as HAdV-F41 for Human Adenovirus species F strain 41. I think it would be worthwhile to use this nomenclature in this text.

Response: We are deeply thankful for this important note and accordingly we updated the nomenclature of adenovirus 41 and adopted the use of “HAdV-F41” throughout the manuscript.

  1. The color designations in Fig 2 in the UK region do not match the corresponding colors in this region in Figs 1 and 3. This should be corrected.

Response: We would like to thank the reviewer for this important remark. We updated the figures in the revised highlighted manuscript.

Reviewer 3 Report

The work titled “Clusters of Hepatitis of Unknown Origin and Etiology (Acute Non HepA–E Hepatitis) Among Children in 2021/2022: A Review of the Current Findings” contributes updated information about the global distribution of this pathology; and the risk this represents for the child population if its causes cannot be determined. Several hypotheses of the possible etiology are addressed in a clear way in this work. Moreover, the authors highlight the need the specific guidelines for identification of further cases (particularly in low-income settings) where testing for some viruses is not considered routinely, or the way to do it is not correct due to the low viral load present in the biological sample. In general, this work is understandable and easy to read. The references are so recent that they include many web pages of state and international health organizations and some newspapers.

Specific comments

In the lines 56-56, the authors cited the reference 4 when they say “out of 108 cases that were confirmed in the U.K. up till 12 April 2022, eight cases required liver transplantation”. However, in this reference the case number is different. I checked the same version (28 April 2022) that the authors mention in reference section.

In the lines 88 - 112, the authors define confirmed, probable and epi-linked case according to ECDC/WHO and ECDC/UK Health Security Agency. I think that a table with this information would be helpful to see differences in definition of these organizations.

Line 120, “… followed by Scotland (n=13),....” the case number for Scotland is 14 according to reference 8 (version 25 April 2022).

Author Response

Reviewer #3 Comments and Suggestions for Authors

The work titled “Clusters of Hepatitis of Unknown Origin and Etiology (Acute Non HepA–E Hepatitis) Among Children in 2021/2022: A Review of the Current Findings” contributes updated information about the global distribution of this pathology; and the risk this represents for the child population if its causes cannot be determined. Several hypotheses of the possible etiology are addressed in a clear way in this work. Moreover, the authors highlight the need the specific guidelines for identification of further cases (particularly in low-income settings) where testing for some viruses is not considered routinely, or the way to do it is not correct due to the low viral load present in the biological sample. In general, this work is understandable and easy to read. The references are so recent that they include many web pages of state and international health organizations and some newspapers.

Response: We are deeply thankful for the insightful summary of the manuscript topic and positive critical appraisal of this review.

Specific comments

  1. In the lines 56-56, the authors cited the reference 4 when they say “out of 108 cases that were confirmed in the U.K. up till 12 April 2022, eight cases required liver transplantation”. However, in this reference the case number is different. I checked the same version (28 April 2022) that the authors mention in reference section.

Response: We are deeply thankful for this meticulous point. We have to point to the rapid evolution of this emerging issue with continuous and rapid updates of the number of cases, severe cases that required or received liver transplantation and mortalities. Hence, we revised this statement based on the latest estimates by the ECDC as follows: “For example, out of 163 cases that were confirmed in the U.K. up till 3 May 2022, eleven cases required liver transplantation [9]. Similarly, two out of nine cases in Alabama, U.S. required the similar management option for severe hepatitis [3,7]. Moreover, several cases of mortality were attributed to acute non HepA-E hepatitis in the U.S. (five deaths), Indonesia (three deaths), Ireland and Palestine (one death in each), which display the threatening potential of this condition [6,10-14].” With updated list of references.

  1. In the lines 88 - 112, the authors define confirmed, probable and epi-linked case according to ECDC/WHO and ECDC/UK Health Security Agency. I think that a table with this information would be helpful to see differences in definition of these organizations.

Response: We are deeply thankful for this important point. Accordingly, we added a table to summarize the case definitions of cases based on the WHO, ECDC and UK Health Security Agency guidelines. Please check Table 1 in the revised highlighted manuscript.

  1. Line 120, “… followed by Scotland (n=13),....” the case number for Scotland is 14 according to reference 8 (version 25 April 2022).

Response: We would like to thank the reviewer for this comment and accordingly we updated the figure based on the latest estimates.

Round 2

Reviewer 1 Report

Lines 165-170: Please double-check the figure legend of the revised Figure 2.

Author Response

Reviewer #1 Comments and Suggestions for Authors

Lines 165-170: Please double-check the figure legend of the revised Figure 2.

Response: We are deeply thankful for the this note, and accordingly we double-checked the figure 2 and its legend.